# School Principals’ Work Participation in an Extended Working Life—Are They Able to, and Do They Want to? A Quantitative Study of the Work Situation

**DOI:** 10.3390/ijerph19073983

**Published:** 2022-03-27

**Authors:** Kerstin Nilsson, Anna Oudin, Inger Arvidsson, Carita Håkansson, Kai Österberg, Ulf Leo, Roger Persson

**Affiliations:** 1Division of Occupational and Environmental Medicine, Lund University, SE-223 81 Lund, Sweden; anna.oudin@med.lu.se (A.O.); inger.arvidsson@med.lu.se (I.A.); carita.hakansson@med.lu.se (C.H.); roger.persson@psy.lu.se (R.P.); 2Department of Public Health, Kristianstad University, SE-291 88 Kristianstad, Sweden; 3Department of Public Health and Clinical Medicine, Umeå University, SE-901 87 Umeå, Sweden; 4Department of Psychology, Lund University, SE-221 00 Lund, Sweden; kai.osterberg@psy.lu.se; 5Department of Centre for Principal Development, Umeå University, SE-901 87 Umeå, Sweden; ulf.leo@umu.se

**Keywords:** principals, organisation, mental, psychosocial, burnout, work environment, health, work ability, work engagement, swage-model, older workers, senior employees

## Abstract

The objective of this study is to increase the knowledge regarding school principals’ work situations by examining the associations between various factors and the school principals’ assessments of their ability or wish to work until 65 years of age or longer. The 1356 participating school principals in this study were aged between 50 and 64 years of age. Individual and work factors were evaluated in relation to two dichotomized outcomes: i.e., can work and want to work beyond 65 years of age, respectively. Generalized Estimating Equations (GEE) models were used to specify bivariate and multivariate cross-sectional logistic regression models that accounted for repeated measurements. The results showed that, both in 2018 and 2019, about 83% of the school principals stated that they could work and about 50% stated that they wanted to work until 65 years of age and beyond. School principals’ exhaustion symptoms and experiences of an excessive burden were statistically significantly associated with whether they both could not and did not want to work beyond 65 years of age. Additionally, the school principals’ experiences of support from the executive management in the performance of their managerial duties was of primary importance for whether the school principals wanted to work until 65 years of age and beyond. To conclude, it is important that school principals receive sufficient support from the management to cope with their often very stressful leadership tasks so that they have the opportunity to be able and willing to continue working their entire working life. The study strengthens the robustness of the theoretical SwAge model regarding the investigated factors related to determinant factors for a sustainable working life and as a basis for developing practical tools for increased employability for people of older ages.

## 1. Introduction

The increasing number of older citizens in society is widely seen as one of the most significant threats to global wealth, due to the tensions that impact the welfare systems through potentially profound economic consequences and social effects [1]. In 2080, it is estimated that for every person of working age (20–64 years), there will be 0.6 people who are 65 years of age or older in the countries that are part of the Organisation for Economic Cooperation and Development (OECD) [1]. This means that every pensioner (if the retirement is set to age 65 years of age) will need to be supported by fewer than two people active in the labour force. This can be compared, for example, with the fact that in the Swedish pension system in 1950, six people in the labour force supported every pensioner [1]. In order to achieve sufficient financial sustainability and to maintain pension systems, many countries are postponing the retirement age in order to make a larger amount of people work for an extended working life and thus contribute to the national economy. Accordingly, there are both societal and organisational interests in increasing the knowledge concerning people’s beliefs and assessments regarding whether they can extend their working life, as well as if they want to extend their working life. Presumably, this knowledge will facilitate the possibilities of taking appropriate organisational measures that will increase the possibility of an extended working life [2,3,4,5,6,7].

In Sweden, school principals have a key role in society with responsibilities entailing, for example, the implementation of governmental directives and the responsibility for teachers’ and pupils’ local work environments as well as educational development (i.e., the responsibility for production, operation and organisation) [8,9]. However, and even if research on principals’ work environments is limited, recent Swedish research has shown that role conflicts, resource deficits and harboring co-workers’ frustrations are demands that many principals frequently face in their managerial position [10]. According to the Teaching and Learning International survey from 2018, principals in Sweden are particularly stressed by excessive administrative tasks, understaffing and working with students with special needs [11]. It has been estimated that approximately 25% of the principals in Sweden are at risk of developing poor mental health if their situation becomes long-lasting [12]. In addition, principal turnover is considered a problem in Sweden—it has been estimated that approximately one in four principals switched schools in the semesters 2011/12, 2012/13 and 2014/15 [13]. Indeed, a high turnover rate may disrupt school development, affect student learning and require the school to recruit new people to fill vacant leadership positions. Nevertheless, compared to many other countries, the cause for principal turnover is less studied in a Swedish context, and turnover rates may differ between age groups, school level and type of municipality (e.g., urban or rural) [14]. Like turnover, retirement also risks disrupting school development, creating a need to fill vacant leadership positions and impacting student learning and the working conditions of staff at the school. However, to date, no study seems to have examined the factors and circumstances associated with principals’ withdrawal from their working life—in other words, their retirement.

In the case of high withdrawal from workplaces, measures need to be taken. It is therefore important to detect problems in the work situation and identify what needs to be improved to support healthy and sustainable employability. The SwAge model is a theoretical model that acknowledges areas of employability and states nine impact and determinant areas connected to sustainable healthy working life and whether individuals can and want to work or not [2,3,4,5,6,7]. Those nine determinant areas are sorted into four spheres based on research on individuals’ consideration of leaving the workplace or being able to stay:⮚The health effects of the work environment, which include the following areas of determination: (1)Self-rated health, diagnoses and diverse physical and mental health functionalities in work, in relation to(2)Physical work environment and work accidents;(3)Mental work environment with risk of stress and fatigue syndrome, threats of violence; and(4)Working hours, work pace and recuperation during and between work shifts.

Working life affects biological aging, and adequate health is a prerequisite for employability and to be included in working life [2,3,4,5,6,7].

⮚Financial incentives for organisations and workplace finances determine which equipment and techniques can be used to facilitate a more sustainable work environment, thus increasing long-term employability and the staffing ratio, as well as sick leave, unemployment and early retirement, not least in bad times.(5)The effects of personal financial situations on issues regarding individuals’ needs and willingness to work due to the possibility of working in relation to the individual employees’ health situation, skills, etcetera, affecting their employability [2,3,4,5,6,7].⮚Relationships and participation concern whether the individual receives sufficient social support from the environment when needed or is included or excluded in the group due to attitudes in the social context. The sphere includes the following areas of determination: (6)The effects of the personal social environment, with a family, friends and leisure context; and(7)The social work environment with leadership, discrimination and the significance of the employment relationship context for individuals’ work.

The relationships and social supports relate to the individuals’ social age and affect the individuals’ opportunities and willingness to work [2,3,4,5,6,7].

⮚Execution of tasks and activities at work can be a source of motivation, stimulation and joy, but it can also be a source of boredom, dissatisfaction and stagnation. Employability also concerns the employees’ ability to meet the requirements in terms of knowledge and skills to execute the activities and tasks and depends also on instrumental support [2,3,4,5,6,7]. The sphere includes the following areas of determination:(8)Motivation, appreciation, satisfaction and stimulation in the execution of work tasks; and(9)Knowledge, competence and the importance of competence development for the individual’s work.

The SwAge model is based on and was developed through the grounded theory method and is based on quantitative and qualitative empirical studies, register data and literature reviews regarding factors of employability and the importance of a healthy and sustainable working life for all ages. The SwAge model also addresses areas of importance for both being able and wanting to work and choosing to leave or stay at a workplace. In addition, the SwAge model also describes how aging relates to various factors in the work environment and work situation. For this reason, the SwAge model is used as a reference framework in the discussion of the analysis results in this study.

### Aim

The objective of this study was to increase the knowledge regarding school principals’ work situations by examining the associations between various individual factors and factors in the work situation and the school principals’ assessment of their ability or wish to work until 65 years of age or longer. Specifically, we asked the following research questions:To what extent do Swedish school principals report that they can and want to work beyond 65 years of age?How strongly are indicators for (a) exhaustion symptoms; (b) mental resources, work engagement and work ability as well as (c) supportive and demanding managerial circumstances related to principals’ statements of whether they (a) can and (b) want to work until 65 years of age or beyond.

Presumably, this knowledge could inform various stakeholders (e.g., supervisors and politicians) and help to tailor initiatives that allow principals to extend their working life in a gainful way and thus prolong their possibilities of contributing to society.

## 2. Materials and Methods

### 2.1. Study Design

The present explorative longitudinal study entailed two assessments one year apart: in the autumn of 2018 and in the autumn of 2019. The data were collected through a web survey. The participants were recruited from a list of e-mail addresses including participants from the compulsory principal education and training programs during the period of 2008–2017, funded and arranged by the Swedish National Agency for Education. The e-mail list had a nationwide reach and entailed 9900 e-mail addresses. Of the 9900 invited, 4640 potential respondents either accepted (*n* = 2 633) or declined (*n* = 2007) to participate. In the end, 2317 respondents completed the entire questionnaire of the first survey in 2018 (i.e., a 50% response rate among those who actively responded yes or no to participation). One year after the first survey, in the autumn of 2019, all except one retiree were once again invited to participate in the second part of the study. Of the 2316 re-invited individuals, 202 could not be reached through their e-mail address anymore (i.e., terminated employment, etc.), 42 refrained from participation, and 544 did not respond to the second survey (66% response rate among those who participated in 2018). In addition, 468 principals participated in 2019, but not in 2018. In total, 1992 principals participated in 2019, and 2781 principals participated in either 2018, 2019 or both. For the analyses carried out in the present paper, only participants aged 50 to 64 years of age at the time of the questionnaire (2018 or 2019) were selected, ending up in a final study sample consisting of 1356 principals (1102 women, 250 men and 4 participants who did not disclose their gender). In this sample, 342 persons participated only in 2018, 307 persons participated only in 2019 and 707 persons participated in both surveys.

### 2.2. Outcome Measures

The outcome measures constituted two single-item measures that have been used in previous studies investigating factors associated to retirement, public health and a sustainable working life until an older age [15,16,17,18,19,20,21]. The first item read “My experience is that I CAN work in my profession until age …”. The second item read “I WANT to withdraw from my professional work when I am… years of age”. Both items were responded to using an eight-step categorical scale indicating the expected age for one’s own retirement in years: 55–60, 61–62, 63–64, 65, 66–67, 68–69, 70–72 and 73 years or older. The response options were dichotomized at 65 years of age (i.e., working until <65 versus ≥65 years of age). The reason for this is because 65 years is the most common retirement age in Sweden, as in many other countries [1], and to withdraw from working life before this age is perceived as an early retirement age.

### 2.3. Stress and Exhaustion Symptoms

The Lund University Checklist for Incipient Exhaustion (LUCIE) was used to measure early manifestations of stress and exhaustion [22,23]. The 28 items of the checklist cover 6 dimensions: sleep and recovery (3 items), separation between work and leisure time (4 items), sense of community and support in the workplace (2 items), managing work duties and personal capabilities (5 items), personal life and leisure time activities (3 items) and health complaints (11 items) [22,23]. The items are responded to on a four-point scale: 1 = not at all, 2= somewhat, 3 = quite a bit, and 4 = very much. The scoring builds two separate but supplementary scales: the Stress Warning Scale (SWS) and the Exhaustion Warning Scale (EWS). Specifically, SWS is sensitive to milder signs of incipient exhaustion, whereas the Exhaustion Warning Scale (EWS) reflects more severe signs of exhaustion. The SWS and EWS are in practice combined into a four-step ranking of incremental stress symptomatology, with the highest level possibly being indicative of ED: Step 1—GG (SWS green zone and EWS green zone), Step 2—YG (SWS yellow zone and EWS green zone), Step 3—RG (SWS red zone and EWS green zone), and Step 4—RR (SWS red zone and EWS red zone). Thus, increasing scores reflect a larger amount of stress symptoms. In the present study, low levels of indications and warnings of demanding stress and exhaustion were used as a reference (Step 1) versus moderate to high (Steps 2–4) indications and warnings of demanding stress and exhaustion symptoms.

Additionally, the Karolinska Exhaustion Disorder Scale (KEDS) was used to assess exhaustion symptoms [24]. KEDS comprises nine items regarding ability to concentrate, memory, physical stamina, mental stamina, recovery, sleep, hypersensitivity to sensory impressions, experience of demands and irritation and anger (24). Each item is responded to on a seven-point scale (0–6). Short descriptive verbal phrases serve as anchors for the scale steps 0, 2, 4 and 6, but not for 1, 3 and 5. Higher scores reflect a more severe exhaustion symptomatology. In the present study, the mean KEDS sum score (range 0 to 54) was used as an outcome. A score ≥19 indicates possible exhaustion disorder [24]. 

### 2.4. Personal Mental Resources

Mental resources were assessed with three items from the Work Ability Index (WAI) [25,26]. The three items were WAI 7a. Have not been able to enjoy daily activities lately; WAI 7b. Have not felt alert and spirited lately; WAI 7c. Have not felt hopeful for the future lately. The items were responded to on a five-point scale: 1 = often, 2 = fairly often, 3 = sometimes, 4 = seldom, 5 = never. For the purposes of analysis, each item was dichotomized into 0 = sometimes, seldom and never and 1 = often and fairly often and used individually in the statistical analyses.

### 2.5. Work Ability

Items from the Work Ability Index (WAI) [25] were used to measure the school principals’ perceptions of their work ability. The used items were as follows: WAI 1a was used to measure the optimistic assessment of the current work ability in comparison to the lifetime best work ability and was responded to with an 11-step Likert type scale with verbal anchors at the endpoints (0 = completely unable to work and 10 = work ability at its best). In addition, WAI 2a was used to measure poor work ability in comparison with the physical needs in the work environment, and WAI 2b was used to measure poor work ability in comparison to the mental needs in the work environment. Both WAI2 items were responded to on a five point scale: 1 = very good, 2 = rather good, 3 = moderate, 4 = rather poor, 5 = very poor. For the purposes of analysis, each item was dichotomized into 0 = moderate, rather poor, very poor and 1 = rather good and very good and used individually in the statistical analyses. Finally, WAI 6 was used to measure whether health is too poor to cope with the current profession in two years from now, based on present health status, and was responded to with a three-step scale: 1 = yes, relatively certain, 2 = not certain and 3 = no, it is unlikely. For the purposes of analysis, the was dichotomized into 0 = yes, relatively certain and 1 = not certain and no it is unlikely.

### 2.6. Work Engagement

The Utrecht Work Engagement Scale comprises nine items (UWES-9) [26]. UWES-9 entail three aspects of engagement: vigor (experience of good energy and enthusiasm for the work), dedication (experience of happiness and inspiration for the work) and absorption (experience of flow in work and pride of work performance). Each aspect encompass three items and responses are made on a seven-point scale (0–6): 0 = never, 1 = almost never, a few times a year or less, 2 = rarely, once a month or less, 3 = sometimes, a few times a month, 4 = often, once a week, 5 = very often, a few times a week and 6 = always, every day. Following the manual, results are reported as the mean value of each aspect. Higher scores indicate increasingly frequent experiences of work engagement. 

### 2.7. Demanding and Supportive Managerial Circumstances

A brief version of the Gothenburg Manager Stress Inventory (GMSI) [27] was used to assess demanding and supportive managerial circumstances. The brief version includes 32 items. Demanding managerial circumstances encompass 22 items covering eight organisational areas: resource deficits, organisational control deficits, role conflicts, role demands, group dynamics, buffer function, co-workers and container function. The items were responded to using a five-point scale: 1 = never, almost never, 2 = rarely, 3 = sometimes, 4 = often, 5 = always, almost always. Supportive managerial circumstances encompass 10 items and cover four organisational areas as well as one area concerning personal life: supportive management, co-operation with co-workers, supportive leader colleagues, supportive organisational structures and supportive personal life. The items were responded to on a five-point scale indicating the participant’s level of agreement: 1 = applies very poorly, 2 = applies poorly, 3 = applies to some extent, 4 = applies well and 5 = applies very well. For both demanding and supportive factors, the mean score was used as an indicator. A higher mean score indicates experiencing demands more frequently or perceiving less support.

### 2.8. Statistical Analysis

To investigate the association between the items, Generalized Estimating Equations (GEE) models were used to specify a repeated measures model with two different dichotomous outcome variables (binary logistic regression) and multiple independent factors. GEE:s were chosen because parameter estimates are, under mild regularity conditions, consistent even when the covariance structure is mis-specified, which make them attractive to use in the present setting. The first dichotomous outcome comprised the item that assessed whether the school principals stated they were able to work until 65 years of age or not. The second dichotomous outcome comprised the item that assessed whether the school principals reported they wanted to work until 65 years of age or beyond or not.

Data from the initial 2018 survey were combined with data from the second survey in 2019, considering dependent observations since many study subjects had participated in both surveys. Data are presented as odds ratios (ORs) with their 95% confidence intervals (CIs), meaning that *p*-values < 0.05 were considered as statistically significant. Univariate analysis was estimated for all analyses to investigate the association between the independent and the dependent variables. There was an interest in analysing possible demanding and supporting managerial circumstances regarding whether principals’ assessed that they can or want to work until 65 years of age and beyond. To cater to this, multivariate analyses were performed to investigate which items were mostly statistically significantly associated to whether the school principals stated they could or wanted to work in an extended working life. Additionally, the analyses were stratified by sex. First, the univariate analyses were executed, and independent factors with *p*-values < 0.1 in the main analyses (men and women) were selected for a multivariate analysis in the GMSI analysis [28] and entered at the same time. All statistical analyses were performed using SPSS version 25.0 software.

## 3. Results

### 3.1. Occurrence of Statements concerning the Ability and Wish to Work beyond 65 Years of Age

A larger proportion of school principals stated they could work until 65 years of age or beyond when compared with those who stated that they wanted to work until 65 years of age or beyond (Table 1). There were no major differences between the years 2018 and 2019.

The next step of the analysis was to investigate factors associated with why school principals assessed that they were able to and wanted to work until 65 years of age and beyond.

### 3.2. Associations between Principals’ Assessments regarding Whether They Believe They Would Not Be Able to or Would Not Want to Work until 65 Years of Age or beyond, and Their Experience of Mental Health and Exhaustion Symptoms

The findings showed a statistically significant association between school principals’ elevated risk of exhaustion disorder and whether they stated that they were not able to and did not want to keep working beyond 65 years of age (Table 2). Furthermore, there was a statistically significant association between whether the principals reported that they were able to and wanted to work until 65 years of age and beyond and whether their health was too poor to cope with their current profession, as well as whether they experienced a lack of work ability. When separate analyses were carried out among men and women, the variables regarding indications and warnings of demanding stress and exhaustion proved to be statistically significant and negatively associated with whether female principals were able to work for an extended working life (i.e., exhaustion symptoms, not having been able to enjoy daily activities lately, not having felt alert and spirited lately, not having felt hopeful for the future lately, perceiving poor work ability in comparison to the physical and mental needs in the work environment). However, for male principals, the only statistically significant association was with the variable “poor work ability in comparison to the needs in the work environment”. However, except for men regarding the variable “poor work ability in comparison to the physical needs in the work environment”, both sexes showed statistically significant associations with all the variables that indicated demanding stress and exhaustion symptoms and whether they did not want to work until 65 years of age. 

### 3.3. Associations between School Principals’ Assessment regarding Whether They Cannot or Do Not Want to Work until 65 Years of Age or beyond and Their Mental Resources, Work Engagement and Work Ability 

The logistic regression results regarding whether school principals experience a good energy and enthusiasm for work, their experience of happiness and inspiration in work and their experience of flow in work and pride of their work performance showed a statistically significant association with whether school principals could and wanted to work until 65 years of age or beyond for both men and women (Table 3). Stratification by sex concerning the same question proved statistically significant associations for almost all variables. However, an optimistic assessment of the current work ability in comparison to the lifetime best work ability was statistically significant only for whether female school principals experienced that they could work for an extended working life.

### 3.4. Associations between School Principals’ Assessments regarding Whether They Believe They Could Not or Did Not Want to Work until 65 Years of Age or beyond and Their Experience of Demanding and Supporting Managerial Circumstances

In order to measure demands and support in working life and the association with whether the school principals thought that they were able to or wanted to remain working for an extended working life until or beyond 65 years of age, we used the GMSI questionnaire. The univariate estimates in the binary logistic regression analyses with repeated measurements showed statistically significant associations between whether the school principals in the total group stated that they were able to work until 65 years of age or beyond and the following areas of analysis: demanding role requirements, container function, supportive management, supportive organisational resources, buffer issues, resource imbalance, organisational deficiencies, logic conflicts, supportive colleagues and a supportive personal life (Table 4).

However, the multivariate modelling only proved a statistically significant association (*p* < 0.05) for 1 of 13 variables. In particular, role demands—that is, burdensome role requirements (i.e., the experience of the burden in terms of the responsibility to be a manager)—proved statistically significant and associated (OR 1.43) with whether the school principals experienced that they could work for an extended working lif, and was more likely for female school principals (OR 1.44). However, this relationship proved not to be statistically significant for male school principals when stratified for sex. Furthermore, the multivariate model proved that supportive colleagues—i.e., support from managers’ colleagues in performing the managerial duties (OR 1.04)—was found to be nearly statistically significantly (*p* < 0.06) associated with whether school principals were able to work for an extended working life. 

In the analysis of whether various areas in the school principals’ work situations were associated with wanting to work for an extended working life, the univariate estimates of the areas of demanding role requirements, group dynamic issues, container function, supportive management, supportive colleagues, supportive organisational resources, buffer issues, resource imbalance, organisational deficiencies and logic conflicts all showed statistically significant associations (Table 4). The multivariate modelling finally stated that 2 of 13 investigated variables were found to be statistically significant and associated with whether the school principals wanted to work for an extended working life. In particular, role demands—that is, burdensome role requirements—proved to be statistically significant and associated with whether the school principal wanted to work until 65 years of age or beyond (in the total group: OR 1.44; women: OR 1.35, men: OR 1.52).

## 4. Discussion

Due to the demographic challenge, a larger number of people need to extend their working life [1]. Furthermore, it takes time to build leadership experiences, and there are too few individuals with the competence of being a school principal in society; therefore, we need to retain competence and experience within organisations [8,9]. However, there is a difference between being able to execute work tasks and wanting to do so— i.e., whether individuals can and/or want to work for an extended working life—as shown by earlier research and stated in the theoretical SwAge model regarding a sustainable working life for all ages [2,3,4,5,6,7,14,15,16,17,18,29]. In this study, the results showed that 83.5% of the school principals stated that they were able to work until 65 years of age and beyond, though only 51.0% wanted to keep working until 65 years of age and beyond. This could be compared to another study, conducted in 2018, which stated that 72.6% of Swedish teachers (*n* = 682) thought they were able to work until 65 years of age and beyond, though only 43.0% of the participants wanted to keep working until 65 years of age and beyond [19]. Additionally, according to another study with Swedish managers in different sectors (except school principals), 86.3% of the included managers (*n* = 153) stated that they were able to work until 65 years of age and beyond, though only 39.9% of the participants wanted to keep working until 65 years of age and beyond [18]. Despite the difference between whether the school principals stated that they were able to and whether they wanted to work in an extended working life, their willingness to remain working proved to be higher than other professions in the same occupational area [18]. 

### 4.1. School Principals’ Risk of Exhaustion Disorders and Poor Health That Affect Their Possibility of Being Able to and Wanting to Work for an Extended Working Life

The results of this study suggest that school principals with signs of stress and exhaustion and risk of developing mental health problems related to their work situation showed an increased risk of assessing that they were not able to or wanted to keep working for an extended working life, especially for female school principals. In previous research, the school principal profession has been described as a risk group regarding the stated connections between work environment and stress-related mental disorders; e.g., depression, anxiety and fatigue [10,11]. Today, the school principal profession is female-dominated, with about 70% female school principals in Sweden. The school principal is controlled and based on political decisions, although the work is executed closest to the citizen, and the role is often caught between these two aspects in their duties [30,31]. Additionally, many school principals in Sweden have reached a mature age, and many will withdraw from working life for retirement within some years [32]. It is apparent that many school principals today, in their roles as leaders, must deal with the consequences of several school reforms that have resulted in major changes in the governance of schools. Moreover, school principals often work based on very varied conditions in terms of organisational and operational responsibilities [9,33]. Work tasks require school principals to manage structural problems (e.g., insufficient resources and organisational governance deficiencies), value conflicts (e.g., different values and norm systems that may collide) and staff problems (e.g., sick leave and staffing) [34]. The work situation of school principals also risks having secondary effects on others, such as impacts on the employees’ work environment and on students’ work environment as well as educational results [35]. Therefore, it is important to make the school principals’ work situation more sustainable through health-promoting organisational strategies and policies, due to the influence and possible effects on many other aspects and people in society. For example, previous research and the SwAge model state that health, personal financial situation and social environments have a profound impact on the individuals’ attitudes towards their ability and willingness to participate in an extended working life or not [2,3,4,5,6,7]. Additionally, the individual’s physical work environment, mental work environment, work schedule, work pace and possibility of recuperation, as well as knowledge, skills and competence development in relation to work demands might be key determinants for whether an individual decides that they can remain working for an extended working life or not. Furthermore, the workplace social environment and the experience of motivation, stimulation and meaningfulness in the work tasks seem to be major determinant areas that affect an individual’s willingness to keep working in an extended working life or not [2,3,4,5,6,7].

### 4.2. The School Principals’ Strategies to Cope with Their Work Situation in Association with Whether They Can and Want to Work for an Extended Working Life

Most of the principals participating in this study reported a fair work engagement and work ability, which proved to be statistically significant and associated with whether they assessed that they were able to and wanted to work for an extended working life. However, in an earlier study based on the current dataset, which also entailed principals below 50 years of age, we observed that approximately 25% of the participating principals seemed to be in a state of mental distress according to their ratings in two inventories that are used in clinical practice: LUCIE and KEDS. [12]. If measures are not taken to counteract further declines in health, this could risk decreasing their capacity to work, their employability and ability to function as school principals, consequently forcing them into early retirement. In previous studies, work ability has been suggested to be a good predictor for future absence due to sickness and early retirement [36,37]. An optimistic assessment of the current work ability in comparison to the lifetime best work ability proved a statistically significant association with wanting to work in an extended working life in this study. However, that association only proved to be statistically significant for women and not for men. The possibility to remain employable and work is a complex issue. The theoretical SwAge model (sustainable working life for all ages) states that employability and whether people can and want to work depend on four spheres: A. the health impacts of the work environment, including the determinant areas of (1) self-rated health, diagnoses and functional diversity; (2) physical work environment and prevention of injuries; (3) mental work environment, stress, effort/reward balance, violence and threats and (4) working hours, work pace and time for recuperation; B. financial incentives, including the determinant area of (5) personal finance; C. social support and participation, including the determinant areas of (6) personal social environment and (7) social work environment; D. self-fulfilment through work tasks, including the determinant areas of (8) motivation, stimulation and self-crediting tasks, the core of work and work satisfaction and (9) competence, skills, knowledge and opportunities for development [2,3,4,5,6,7,29,38,39]. Regardless of profession, all nine determinant areas based on the theories in the SwAge model are important to examine in order to promote an individual’s assessments of whether they can and want to keep working in an extended working life. Additionally, management tools and measurement activities may be needed to increase the school principals’ abilities and wishes to extend their working life.

### 4.3. The School Principals’ Vulnerability in Their Tasks and Roles as Principals in Association with Whether They Can and Want to Work for an Extended Working Life 

The multivariate modelling in this study regarding whether school principals’ experiences that they could work for an extended working life proved statistically significant associations with their experience of burdens in terms of the responsibilities in the role of a manager, as well as the support from the managers’ colleagues in their performance of their managerial duties. The role of a school principal includes the need for skills, knowledge and competence development to cope with and execute the work tasks as a manager and in the role of a leader. Otherwise, if school principals do not have adequate or full knowledge of how to cope with their role as a manager, the role of a school principal, with a great deal of responsibility, will be mentally demanding and may make the school principals’ personal stress harder to cope with and control. Furthermore, previous research into other professions and the SwAge model state that both adequate knowledge to manage work tasks and the mental work environment, with a balance of demand and control, are important for individuals’ assessments of whether they can work for an extended working life or not [2,3,4,5,6,7,29,38]. 

The other multivariate modelling in this study—whether school principals wanted to work until 65 years of age or beyond—also proved a statistically significant association with their experience of burdens in terms of the responsibilities of the role of being a manager but was also related to the support from executive management in performing their managerial duties. These results are in accordance with previous studies on other professions, which state the need for support and confirmation from their managers and from the management in order to see their own work effort as an important part and their own role as a part of the organisation’s purpose; otherwise, the work can be perceived as meaningless [2,3,4,5,6,7,29,38]. Social and instrumental support, coherence and inclusion in a group are important determinant factors that affect whether individuals want to stay in the workplace and keep working until an older age, as shown in the SwAge model [2,3,4,5,6,7,14,15,16,17,18,29,38]. 

### 4.4. Limitations

A limitation of this study was that the survey mostly focused on the mental and organisational issues of school principals’ work situations and did not investigate all nine determinant areas for a sustainable working life stated in the SwAge model [2,3,4,5,6,7,14,15,16,17,18,29,38]. However, the difficulties and complexities in the school principals’ work situations are probably, to a large extent, contingent on mental and organisational workloads and to a lesser extent on the physical workload. Another limitation could be that the study used questions from previous studies and did not investigate the school principals’ retirement considerations in depth. However, the questions used in the survey were validated and reliability controlled in previous studies, providing robustness to the present study. Additionally, the opportunities of this dataset, being based on a follow-up study and using the same participating respondents in two consecutive years, increase the robustness of the results. This investigation included Swedish school principals throughout the country, from north to south, including principals working at schools in small villages and large cities and school principals from pre-school to high school and adult school. Although framed in a Swedish context, we presume that school principals in other countries and cultures might share similar challenges in their work situations and perceived health due to the occupational similarity of schools; however, the organisations of school systems could be quite different.

## 5. Conclusions

The results show that circa 17% of Swedish school principals do not perceive that they can work until 65 years of age or beyond, and circa 49% do not want to work until 65 years or beyond. In addition, most indicators that covered exhaustion symptoms, poor health, poor mental resources and low work ability in relation to the physical and mental demands of work were positively associated with reports of not being able to work or not wanting to work until 65 years of age or beyond. In contrast, reporting a high workability in relation to respondents’ lifetime best performance and expressing a higher work engagement decreased the probability of reporting being unable or not wanting to work until or beyond 65 years of age. Furthermore, among the organisational variables assessing demanding and supportive circumstances, the multivariate analysis showed that role demands seem to be the best indicator for reporting not being able to work as well as not wanting to work until 65 years of age or beyond. Collectively, the present results seem to suggest that interventions that aim to improve school principals’ individual health and well-being and strive to reduce their role demands are likely to be beneficial for the principals and thus for the school organisation they manage. However, the results also suggest that it might be worthwhile in future research to address the question of school principals’ withdrawal from the labour market from a more comprehensive theoretical framework such as the SwAge model; i.e., that employability and whether people can and want to work depend on four spheres: the health impacts of the work environment, financial resources, social support and experiences of social participation and self-fulfillment through work tasks. In order to increase the likelihood and capacity to function as a school principal, a work environment with a positive impact on health— that is, with a sense of community in the organisation, trust and social support, acceptance from the employees’ executive management and a generally manageable stress exposure—appears to be an important determinant. This could probably reduce the risk of developing health problems. Furthermore, such interventions may likely increase school principals’ willingness and ability to work for an extended working life. A work situation experienced as healthy, supportive and trustful, as well as stimulating and meaningful for individuals’ professional roles and duties, is likely a good way to inspire school principals to keep working for an extended working life.

## Figures and Tables

**Table 1 ijerph-19-03983-t001:** School principals’ statements regarding whether they “Can work” or “Want to work” until 65 years of age and beyond in 2018 and in 2019. (no = can/want only to work until 55 to 64 years of age; yes = I can/want to work 65 years or beyond).

	School Principals’ Statements			
	Can Work until 65 Years or beyond		Want to Work until 65 Years or beyond	
	No	Yes	No	Yes
2018	16.8%	83.2%	49.5%	50.5%
2019	17.3%	82.7%	48.8%	51.2%

**Table 2 ijerph-19-03983-t002:** Univariate estimates from multiple binary logistic regression with repeated measurements on school principals’ assessment of whether they “cannot work” or “do not want to work” until 65 years of age and beyond in association with the school principals’ experience of health problems in their situation for the total study population and stratified by sex. The results are presented as odds ratios (OR) with accompanying 95% confidence intervals (CI) and *p*-values.

	Outcome Variables	Group	Cannot Work until 65 Years of Age or beyond			Do Not Want to Work until 65 Years of Age or beyond		
Independent Variables			OR	95% CI	*p*-Value	OR	95% CI	*p*-Value
Indications and warnings of demanding stress and exhaustion symptoms (LUCIE)		Total	1.99	1.53–2.59	<0.0001	1.68	1.36–2.07	<0.0001
		Women	1.99	1.50–2.62	<0.0001	1.62	1.28–2.05	<0.0001
		Men	1.88	0.85–4.16	0.12	1.81	1.24–2.65	0.002
Exhaustion disorder indications (KEDS)		Total	1.92	1.51–2.45	<0.0001	1.57	1.29–1.91	<0.0001
		Women	1.88	1.45–2.43	<0.0001	1.47	1.18–1.83	0.0007
		Men	2.02	0.97–4.20	0.06	1.96	1.34–2.86	0.0005
My health is too poor to cope with the current profession, based on my current health status (WAI_6)		Total	6.01	3.70–9.76	<0.0001	2.80	1.72–4.56	<0.0001
		Women	7.66	4.44–13.20	<0.0001	2.79	1.59–4.88	0.0003
		Men	2.20	0.66–7.47	0.20	2.93	1.10–7.79	0.03
Have not been able to enjoy daily activities lately (WAI_7a)		Total	1.96	1.25–3.07	0.003	1.47	1.24–0.45	0.08
		Women	2.03	1.25–3.28	0.004	1.47	0.90–2.38	0.12
		Men	1.68	0.46–6.13	0.44	1.58	0.67–3.73	0.29
Have not felt alert and spirited lately (WAI_7b)		Total	2.75	1.97–3.84	<0.0001	2.13	1.59–2.85	<0.0001
		Women	2.80	1.94–4.02	<0.0001	1.93	1.40–2.65	<0.0001
		Men	2.72	1.13–6.51	0.03	3.55	1.75–7.22	0.0005
Have not felt hopeful for the future lately (WAI_7c)		Total	2.36	1.61–3.47	<0.0001	1.50	1.08–2.08	0.01
		Women	2.38	1.58–3.60	<0.0001	1.31	0.91–1.88	0.15
		Men	2.58	1.01–6.59	0.05	2.93	1.41–6.10	0.004
Poor work ability in comparison to the physical needs in the work environment (WAI_2a)		Total	4.21	2.03–8.74	0.0001	2.60	1.27–5.13	0.01
		Women	4.13	1.90–8.97	0.0003	2.60	0.76–0.87	0.02
		Men	4.37	0.41–37.76	0.23	2.36	0.37–14.90	0.36
Poor work ability in comparison to the mental needs in the work environment (WAI_2b)		Total	3.46	2.38–5.04	<0.0001	2.01	1.46–2.79	<0.0001
		Women	3.36	2.22–5.10	<0.0001	1.97	1.36–2.86	0.0003
		Men	4.35	1.86–10.15	0.0007	2.21	1.18–4.12	0.01

**Table 3 ijerph-19-03983-t003:** Univariate estimates from multiple binary logistic regression with repeated measurements on school principals’ assessment of whether they “cannot work” or “do not want to work” until 65 years of age and beyond in association with the school principals’ experience of enthusiasm (continuous variables), for the total study population and stratified by sex. The results are presented as odds ratios (OsR) per one unit increase in the independent variables with accompanying 95% confidence intervals (CsI) and *p*-values.

	Outcome Variables	Group	Cannot Work until 65 Years of Age or beyond			Do Not Want to Work until 65 Years of Age or beyond		
Independent Variables			OR	95% CI	*p*-Value	OR	95% CI	*p*-Value
Optimistic assessment of the current work ability in comparison to the lifetime best work ability (WAI_1)		Total	0.80	0.75–0.85	<0.0001	0.82	0.78–0.87	<0.0001
		Women	0.77	0.72–0.83	<0.0001	0.81	0.76–0.87	<0.0001
		Men	0.91	0.78–1.06	0.21	0.86	0.78–0.95	0.003
Experience of good energy and enthusiasm for work (UWES_Vigor)		Total	0.61	0.54–0.69	<0.0001	0.69	0.62–0.76	<0.0001
		Women	0.59	0.52–0.68	<0.0001	0.68	0.61–0.77	<0.0001
		Men	0.65	0.46–0.93	<0.0001	0.70	0.57–0.86	0.0006
Experience of happiness and inspiration in work (UWES_Dedication)		Total	0.61	0.52–0.72	<0.0001	0.64	0.56–0.70	<0.0001
		Women	0.59	0.50–0.71	<0.0001	0.63	0.54–0.74	<0.0001
		Men	0.67	0.43–1.03	0.07	0.62	0.47–0.84	0.0016
Experience of flow in work and pride of work performance (UWES_Absorbtion)		Total	0.72	0.63–0.84	<0.0001	0.71	0.62–0.80	<0.0001
		Women	0.59	0.58–0.81	<0.0001	0.71	0.62–0.83	<0.0001
		Men	0.73	0.53–1.00	0.05	0.61	0.63–0.78	0.0001

**Table 4 ijerph-19-03983-t004:** Univariate and multivariate estimates from multiple binary logistic regression with repeated measurements on school principals’ assessments of whether they “cannot work” or “do not want to work” until 65 years of age and beyond in association with demanding and supporting managerial circumstances (GMSI variables), for the total study population and stratified by sex. The results are presented as odds ratios (ORs) with accompanying 95% confidence intervals (CIs) and *p*-values.

Scale Name	Explanation	Group	Cannot Work until 65 Years of Age or beyond						Do Not Want to Work until 65 Years of Age and beyond					
			Univariate Estimates			Multivariate Model			Univariate Estimates			Multivariate Model		
			OR	95% CI	*p*-Value	OR	95% CI	*p*-Value	OR	95% CI	*p*-Value	OR	95% CI	*p*-Value
Demanding scircumstances														
Resource deficits	Resource imbalance, i.e., financial imbalance	Total	1.18	1.06–1.32	0.004	1.03	0.91–1.18	0.6292	1.11	1.02–1.21	0.018	1.03	0.90–1.18	0.6495
		Women	1.19	1.05–1.34	0.0077				1.07	0.97–1.17	0.1936			
		Men	1.15	0.89–1.48	0.2765				1.31	1.08–1.59	0.0052			
Organisational control	Organisational deficiencies, i.e., lack of transparency and communication from management	Total	1.21	1.07–1.37	0.0027	1.02	0.86–1.20	0.9916	1.13	1.03–1.25	0.0145	1.02	0.86–1.20	0.8597
		Women	1.21	1.06–1.39	0.0044				1.11	1.00–1.24	0.0605			
		Men	1.18	0.86–1.62	0.2967				1.20	0.94–1.52	0.1407			
Role conflicts	Logic conflicts, i.e., imbalance between administration and practical management work	Total	1.32	1.15–1.52	<0.0001	1.00	0.83–1.21	0.9916	1.22	1.11–1.36	0.0001	1.01	0.84–1.22	0.9055
		Women	1.30	1.11–1.52	0.0010				1.22	1.09–1.37	0.0007			
		Men	1.37	1.00–1.86	0.0473				1.18	0.95–1.45	0.1292			
Role demands	Burdensome role requirements, i.e., the experience of the burden in terms of the responsibility to be the manager	Total	1.63	1.40–1.89	<0.0001	1.43	1.17–1.75	0.0005	1.48	1.31–1.67	<0.0001	1.44	1.18–1.77	0.0004
		Women	1.60	1.36–1.89	<0.0001	1.44	1.16–1.79	0.0009	1.45	1.26–1.66	<0.0001	1.35	1.13–1.62	0.0008
		Men	1.76	1.17–2.58	0.0064	1.44	0.76–2.75	0.2667	1.57	1.24–2.01	0.0002	1.52	1.09–2.14	0.0146
Group dynamics	Group dynamic issues, i.e., employees’ acceptance of the principal as manager	Total	1.10	0.94–1.28	0.2196				1.08	0.96–1.22	0.2199			
		Women	1.08	0.92–1.27	0.3645				1.06	0.93–1.21	0.4046			
		Men	1.30	0.88–1.91	0.7908				1.14	0.87–1.49	0.3513			
Buffer function	Buffer issues, i.e., being caught between management and employees	Total	1.21	1.06–1.37	0.0039	0.96	0.80–1.14	0.6106	1.18	1.08–1.29	0.0004	0.96	0.80–1.14	0.6432
		Women	1.16	1.01–1.32	0.0302				1.16	1.05–1.28	0.0048			
		Men	1.67	1.11–2.50	0.0136				1.27	1.03–1.56	0.0256			
Co-workers	Employee issues, i.e., supporting employees in their work tasks	Total	1.10	0.94–1.29	0.2464				1.16	1.04–1.30	0.0095	0.89	0.74–1.08	0.2315
		Women	1.01	0.85–1.20	0.9240				1.15	1.01–1.30	0.0339			
		Men	1.65	1.02–2.65	0.0369				1.11	0.89–1.40	0.3604			
Container function	Container function i.e., partly therapeutic role in employees’ (work) stress	Total	1.38	1.18–1.60	<0.0001	1.16	0.96–1.39	0.1272	1.28	1.15–1.43	<0.0001	1.20	0.99–1.47	0.0680
		Women	1.32	1.12–1.56	0.0009				1.28	1.13–1.45	<0.0001			
		Men	1.62	1.09–2.42	0.0180				1.19	0.96–1.48	0.1122			
Supportive circumstances														
Supportive management	Lack of supportive management, i.e., support from executive management to perform the managerial duties	Total	1.16	0.95–1.30	0.0052	1.04	0.91–1.19	0.5485	1.20	1.12–1.32-	<0.0001	1.04	0.91–1.19	0.5636
		Women	1.16	1.04–1.32	0.0090				1.19	1.30–1.87	<0.0001			
		Men	1.20	0.89–1.61	0.2639				1.32	1.10–1.56	0.0029			
Cooperating co-workers	Lack of collaboration with co-workers, i.e., support from the employees to perform managerial duties	Total	1.09	0.92–1.30	0.3308				1.02	0.89–1.15	0.8034			
		Women	1.05	0.88–1.27	0.5661				1.00	0.87–1.15	0.9923			
		Men	1.35	0.80–2.27	0.2609				1.02	0.78–1.34	0.8977			
Supportive (manager) colleagues	Lack of supportive colleagues, i.e., support from managers’ colleague to perform managerial duties	Total	1.16	1.04–1.30	0.0067	1.09	0.96–1.23	0.1824	1.11	1.02–1.21	0.0130	1.09	0.96–1.24	0.102
		Women	1.16	1.04–1.32	0.0089				1.08	0.99–1.18	0.0994			
		Men	1.20	0.84–1.72	0.3123				1.27	1.05–1.54	0.0136			
Supportive personal life	Lack of supportive personal life, i.e., support from family and personal network to perform managerial duties	Total	1.14	0.56–0.94	0.0437	1.02	0.90–1.16	0.7599	1.09	0.99–1.19	0.0960	1.02	0.89–1.16	0.7819
		Women	1.10	0.96–1.25	0.1741				1.06	0.99–1.18	0.0994			
		Men	1.41	0.97–2.04	0.0717				1.14	0.91–1.43	0.2542			
Supportive organisational structures	Lack of supportive organisational resources, i.e., support and legacy in the organisation to perform managerial duties	Total	1.22	1.09–1.37	0.0007	1.06	0.93–1.22	0.3547	1.11	1.00–1.22	0.0560	1.08	0.93–1.23	0.3006
		Women	1.19	1.05–1.35	0.0048				1.06	0.95–1.19	0.2860			
		Men	1.47	1.01–2.08	0.0423				1.22	0.95–1.56	0.1184			

## Data Availability

In accordance with the ethical approval by the Regional Ethical Review Board, crude data are not to be published on the internet. Consistent with the study protocol, anonymised data are stored locally at the Division of Occupational and Environmental Medicine, Lund University, Lund, Sweden. Access to data will be granted to eligible researchers wanting to audit our research. Requests should be directed to the corresponding author.

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
