# Peer review of "School Principals’ Work Participation in an Extended Working Life—Are They Able to, and Do They Want to? A Quantitative Study of the Work Situation"

_ijerph, 2022, doi:10.3390/ijerph19073983_

Round 1
Reviewer 1 Report
This was one of the best and most interesting articles I have reviewed for this journal group. The topic is personally interesting to me as a professor and former principal in the U.S.
My only comment is that i would have liked to have seen more detail describing the "I can work" measurements and a more detailed description there.
I also wondered whether female principals experienced higher levels of stress because they tended to be single parent wage-earners. Breaking down demographics more finely may result in even more findings for this research. Thank you.
Author Response
Comments and Suggestions for Authors Reviewer 1
This was one of the best and most interesting articles I have reviewed for this journal group. The topic is personally interesting to me as a professor and former principal in the U.S.
Response: We thank the reviewer for the positive response.
My only comment is that i would have liked to have seen more detail describing the "I can work" measurements and a more detailed description there.
Response: We agree. Accordingly, we have now clarified our description of our two outcome measures.
I also wondered whether female principals experienced higher levels of stress because they tended to be single parent wage-earners. Breaking down demographics more finely may result in even more findings for this research. Thank you.
Response: The speculations regarding whether female principals are more stressed because they are more frequently are single-wage earners, and whether it is possible to break down demographics even further, are relevant. Yet, we believe that these questions are beyond the scope of the present study and warrant an investigation in their own right. However, it may briefly be noted that we in a previous study have observed that female principals to a somewhat lesser extent reported co-habiting with a partner. However, and while female principals indeed reported more often living apart, an equal amount of male and female principals reported not to co-habit with anyone (3.1% and 3.9 % single, respectively). Accordingly, female principals do not seem to be single-wage earners to a higher extent than male principals. And there was no difference between male and female principals as regards their reporting whether they during the last year had experienced stress and/or pressure caused by factors that were unrelated to their work.
Reviewer 2 Report
This topic is very interesting and novel. The study has important practical implications. However, I also have the following suggestions.
- The manuscript should be more explicit about the theoretical model and hypothesis of the study. This can help the readers to better understand and grasp the content of the study.
- The authors should have explained more about the theoretical contribution of this study. Although, I think the practical implications of this manuscript are obvious. However, the arguments of the theoretical contribution and the advancement of previous literature are not clear enough. What are the theoretical implications of this manuscript?
Author Response
Comments and Suggestions for Authors Reviewer 2
This topic is very interesting and novel. The study has important practical implications. However, I also have the following suggestions.
The manuscript should be more explicit about the theoretical model and hypothesis of the study. This can help the readers to better understand and grasp the content of the study.
Response question 1: We agree and has improved this.
The authors should have explained more about the theoretical contribution of this study. Although, I think the practical implications of this manuscript are obvious. However, the arguments of the theoretical contribution and the advancement of previous literature are not clear enough. What are the theoretical implications of this manuscript?
Response question 2: We agree and has improved this.
Reviewer 3 Report
The main drawbacks of the paper are the following:
The Abstract section should be rewritten. The research ideas could have been more effective using elaborative and concise sentences. The abstract does not provide a concise account of the work and conclusion of the research study. It needs to be more structured and synthesized for research clarity.
The introduction section has one subsection 1.2. Aim. Where is subsection 1.1????
The Literature review or Background section is missing.
Research questions or hypotheses are missing.
The authors should provide a brief description of the methods of data employed and their application and appropriateness for data analysis.
The results were not well-presented to readers to understand the focus of the research study.
The results must be interpretive rather than just descriptive and connect the research results with relevant literature citations for validity and reliability.
The research data does not support the conclusion, which does not indicate a more straightforward path for future studies on the topic.
The literature is limited to 39 sources, which is one of the consequences of a weak dialogue with literature.
Good luck!
Author Response
Comments and Suggestions for Authors (Reviewer 3)
The main drawbacks of the paper are the following:
The Abstract section should be rewritten. The research ideas could have been more effective using elaborative and concise sentences. The abstract does not provide a concise account of the work and conclusion of the research study. It needs to be more structured and synthesized for research clarity.
Response: We agree
The introduction section has one subsection 1.2. Aim. Where is subsection 1.1????
Response: Thank you. This has been rectified. It should stand 1.1
The Literature review or Background section is missing.
Response: We are not sure what the reviewer asks for. In the introduction section we account for the background. And, in line with the journal instructions, we have in the introduction tried to briefly place the study in a broad context and highlight its importance. We also think the purpose and the potential significance of our work is made clear. In addition, and because this is a novel area of research, and a novel study, there are not many original studies (if any) on this topic to draw on. It is also unclear whether the reviewer think we have missed some key-publications.
Research questions or hypotheses are missing.
Response: We do not agree. The objective and the specific questions are clearly presented under heading “1.1 Aims”. However, to avoid confusion we have now improved the phrasing. Specifically, we have strived to make the general objective becomes even more clearly distinguishable from the specific research question.
The authors should provide a brief description of the methods of data employed and their application and appropriateness for data analysis.
Response: We are uncertain what the reviewer asks for. Observably, we provide brief descriptions of our study design, outcome measures, independent variables and how we have statistically analyzed the data. However, in line with the reviewer suggestion, we have now improved the descriptions of our various indicator variables in the methods section. And we have also clarified how we ascertained the appropriateness and assumptions for the statistical test (ANNA?).
The results were not well-presented to readers to understand the focus of the research study.
Response: We agree. Accordingly, we have now read and adjusted the text and tables in the results section.
The results must be interpretive rather than just descriptive and connect the research results with relevant literature citations for validity and reliability.
Response: We do not agree. Although the journal instructions suggest that it is possible to intermingle the results and discussion section, we think the clarity increases by clearly separating the results from the discussion. In addition, irrespective of whether descriptive or interferential statistical methods have been applied, it is also a common practice to present the uninterpreted results in the results section while making the interpretation and contextualization in the discussion section.
The research data does not support the conclusion, which does not indicate a more straightforward path for future studies on the topic.
Response: We agree. The conclusions are perhaps a bit to distal from the data. This has been rectified.
The literature is limited to 39 sources, which is one of the consequences of a weak dialogue with literature.
Response: It is unclear what the reviewer asks for and whether the reviewer believes that we have missed key literature. We think that the relevant articles and papers are included.
Good luck!
Response: Thanks!
Round 2
Reviewer 2 Report
The paper should have dealt more with how the theory (swAge model) was applied in this study. But right now, the theory section is simply describing the theoretical content, and the study is the study. Especially that part of the theory described in the introduction. So, how does this study fit into that theory? Why was the theory chosen for this paper?
Author Response
Thanks for the suggestions for improvement. The SwAge model is based on and developed through the grounded theory method and based on quantitative and qualitative empirical studies, register data and literature reviews regarding factors for employability and of importance for a healthy and sustainable working life for all ages. The SwAge model also addresses areas of importance for both being able and wanting to work and chooses to leave or stay at a workplace. In addition, the SwAge model also describes how aging relates to various factors in the work environment and work situation. For this reason, the SwAge model is used as a reference framework in the discussion of the analysis results in this study.
We have marked in the text with yellow where we have clarified this
Reviewer 3 Report
Good luck!
Author Response
Thank you!